# Iron Rim Lesions as a Specific and Prognostic Biomarker of Multiple Sclerosis: 3T-Based Susceptibility-Weighted Imaging

**DOI:** 10.3390/diagnostics13111866

**Published:** 2023-05-26

**Authors:** Sooyoung Kim, Eun Kyoung Lee, Chang June Song, Eunhee Sohn

**Affiliations:** 1Department of Neurology, Chungnam National University Hospital, Chungnam National University College of Medicine, Daejeon 35015, Republic of Korea; ksy9911@naver.com; 2Department of Neurology, Chungnam National University Sejong Hospital, Chungnam National University College of Medicine, Daejeon 35015, Republic of Korea; 2021eunklee@cnuh.co.kr; 3Department of Radiology, Chungnam National University Hospital, Chungnam National University College of Medicine, Daejeon 35015, Republic of Korea; cjsong@cnu.ac.kr

**Keywords:** multiple sclerosis, iron, diagnostic imaging, disease progression

## Abstract

This study aimed to identify the clinical significance of iron rim lesions (IRLs) in distinguishing multiple sclerosis (MS) from other central nervous system (CNS) demyelinating diseases, determine the relationship between IRLs and disease severity, and understand the long-term dynamic changes in IRLs in MS. We retrospectively evaluated 76 patients with CNS demyelinating diseases. CNS demyelinating diseases were classified into three groups: MS (n = 30), neuromyelitis optica spectrum disorder (n = 23), and other CNS demyelinating diseases (n = 23). MRI images were obtained using conventional 3T MRI including susceptibility-weighted imaging. Sixteen of 76 patients (21.1%) had IRLs. Of the 16 patients with IRLs, 14 were in the MS group (87.5%), indicating that IRLs were significantly specific for MS. In the MS group, patients with IRLs had a significantly higher number of total WMLs, experienced more frequent recurrence, and were treated more with second-line immunosuppressive agents than were patients without IRLs. In addition to IRLs, T1-blackhole lesions were observed more frequently in the MS group than in the other groups. IRLs are specific for MS and could represent a reliable imaging biomarker to improve the diagnosis of MS. Additionally, the presence of IRLs seems to reflect more severe disease progression in MS.

## 1. Introduction

Although the 2017 McDonald diagnostic criteria [1] have improved the sensitivity of diagnosing multiple sclerosis (MS), distinguishing MS from other MS-mimicking demyelinating diseases remains challenging. There have been no definitive imaging biomarkers for diagnosing MS. Although the central vein sign (CVS) within white matter lesions (WMLs) has been suggested to be highly specific to MS, previous studies have shown that the presence of CVS is not related to the clinical phenotype or disease progression of MS [2,3]. In addition to CVS, the presence of T1-blackhole lesions has been considered a characteristic of MS. However, there has been no obvious definition of lesion identification and quantification of T1-blackhole lesions despite the importance and potential role of T1-blackhole lesions as surrogate markers in MS [4].

Iron rim lesions (IRLs) are defined as WMLs surrounded by a hypointense rim in susceptibility-weighted images (SWIs) from brain magnetic resonance imaging (MRI), and these rim lesions have been suggested to reflect chronic active inflammatory demyelination in MS [5,6]. The hypothesis on the formation of IRLs is that iron-enriched macrophages and microglia gather at the edge of the chronic demyelinating lesions when the myelin or oligodendrocyte are insulted by active inflammatory processes [7]. IRLs have been proposed to be a specific diagnostic marker of MS and could be associated with more brain and cervical cord atrophy in MS [8,9,10]. Furthermore, longitudinal imaging studies have shown that WMLs surrounded by a rim are more likely to expand over time than those without rims [11]. Therefore, identifying the long-term dynamic changes of IRLs would help our understanding of the pathophysiology, disease activity, and prognosis of MS.

To date, the results of studies on the relationship between IRLs and the clinical severity of MS have been controversial. Some recent studies have proposed that IRLs are specific for MS and are a potential prognostic biomarker reflecting early disability progression in MS [3,12]. On the other hand, previous research reported that the Expanded Disability Status Scale (EDSS) score and clinical status relating to relapse did not differ significantly between MS patients with and without IRLs [13]. Until now, many previous studies dealing with IRLs in MS have mostly been conducted by using 7T MRI. However, the use of 7T MRI may not be widely available in clinical practice. In this study, we used conventional 3T MRI, which is widely available in clinical practice, to identify the clinical significance of IRLs in distinguishing MS from other central nervous system (CNS) demyelinating diseases, determine the relationship between IRLs and disease severity, and understand the long-term dynamic changes in IRLs in MS.

## 2. Materials and Methods

### 2.1. Patients and Clinical Data

We retrospectively recruited patients with CNS demyelinating diseases who visited Chungnam National University Hospital between January 2019 and July 2022. The patient eligibility criteria were as follows: (1) age > 20 years, (2) clinical/MRI evidence of CNS involvement with demyelinating lesions, and (3) availability of brain SWIs. The patients were classified into three diagnostic groups: MS, neuromyelitis optica spectrum disorder (NMOSD), and other CNS demyelinating diseases (ODD). Thirty patients were diagnosed with MS based on the 2017 McDonald criteria [1]. The diagnosis of NMOSD was based on the revised diagnostic criteria of Wingerchuk et al. [14]. Among twenty-three NMOSD patients, twenty patients were positive for serum anti-aquaporin-4 (AQP-4) IgG antibodies in a cell-based immunohistochemistry test. The remaining three patients were diagnosed with NMOSD based on previous medical records and brain MRI. Twenty-three patients with ODD had myelin oligodendrocyte glycoprotein (MOG)-associated disorders (n = 2), optic neuritis (n = 5), or idiopathic CNS demyelinating disease (n = 16). Sixteen patients with idiopathic CNS demyelinating disease had no abnormalities in laboratory tests for differentiating CNS demyelinating diseases and no clinical progression of the disease or new lesions on MRI without maintenance treatment. Two of the 23 patients with ODD relapsed, including one with anti-MOG antibodies and one with recurrent optic neuritis.

Clinical variables, including age, sex, symptoms at initial presentation, disease duration, follow-up period, number of recurrences, medical treatment for prevention, and EDSS scores, were evaluated. Symptoms at initial presentation were divided into optic neuritis, transverse myelitis, cerebral symptoms, brainstem symptoms, and myelitis with brain symptoms. Disease duration was defined as the period from the date of symptom onset to the date of the first MRI scan. If the patient received steroid pulse therapy or had one or more new lesions detected on MRI, recurrence was considered. The EDSS scores were assessed whenever an MRI scan was performed. Moreover, data on serum anti-AQP4 antibodies, anti-MOG antibodies, anti-Ro/La antibodies, cerebrospinal fluid (CSF) oligoclonal bands (OCB), and IgG index were collected if performable.

### 2.2. MRI Evaluations

MRI images were obtained using four 3T MRI scanners with 32 channels: Discovery MR 750 W (GE Healthcare, Waukesha, WI, USA), MAGNETOM Skyra (Seimens Medical Solutions, Erlangen, Germany), Ingenia Elition X (Philips Healthcare, The Netherlands), and Achieva (Philips Healthcare).

The SWIs, conventional T1-weighted images (WIs), T2-WIs, fluid attenuation inversion recovery (FLAIR) preparation, and contrast-enhanced T1-WIs were obtained, which are protocols recommended by the MS consortium [15]. MRI scans were performed at the first diagnosis and then annually or when recurrence was suspected. The first MRI scans of all patients were analyzed to compare the CNS demyelinating disease groups. Serial MRI findings were compared in the MS group to observe dynamic changes in the IRLs. All images were independently assessed by three neurologists (Kim S, Lee EK, and Sohn E) and one neuroradiologist (Song CJ). Lesions were counted only when two or more reviewers agreed, and after repeated reviews, the final number of lesions for each subject was determined by consensus among all the raters. We counted the number of IRLs and WMLs. IRLs were defined as hyperintense lesions on FLAIR images and a hyperintense core partially or completely surrounded by a hypointense rim on SWIs [4]. Rim lesions were defined as larger than 3 mm in the long axis on T2-WIs, and infratentorial lesions were excluded [4]. We reviewed the magnitude image, phase image, and minimum intensity projection (mIP) images in the SWI protocol to confirm the presence of IRLs. To determine the loading of the IRLs compared to WMLs, the IRL/WML ratio was calculated. We also analyzed the location of IRLs: cortical/juxtacortical, periventricular, and deep white matter lesions [16]. To analyze the dynamic changes in IRLs in the MS group, we used at least two serial MRI scans of MS patients. Temporal changes were classified into three groups according to the changes in the total number of IRLs over time: expansion, decrease, or no change. The number of WMLs on the FLAIR images was counted. If the long axis of the WML was >2 cm and clumped together, it was defined as a confluent lesion. We also counted the number of T1-blackhole and T1-contrast-enhanced lesions. A T1-blackhole was defined as a hypointense lesion compared with the white matter on T1-WIs and in concordance with a hyperintense lesion on T2-WIs [10,13]. 

### 2.3. Statistical Analysis

All statistical analyses were performed using SPSS software (version 26.0; SPSS Inc., Chicago, IL, USA). Statistics for clinical variables are presented as proportions, means, standard deviations, and ranges. The independent *t*-test, Pearson’s chi-square test, Fisher’s exact test, Mann–Whitney U test, and correlation analysis were performed to compare the statistical significance of the groups. The Kruskal–Wallis H-test was applied to independent samples. Statistical significance was set at *p* < 0.05. 

### 2.4. Ethics Statement

This study was approved by the Institutional Review Board of the Chungnam National University Hospital (approval number:2023-04-058). All the procedures were performed in accordance with the principles of the Declaration of Helsinki. The requirement for informed consent was waived due to the retrospective nature of this study. 

## 3. Results

### 3.1. Patient Baseline Characteristics

A total of 76 patients were included in the study (MS, 30; NMOSD, 23; and ODD, 23). Of the 30 MS patients, 28 were relapse-remitting MS (RRMS), one was diagnosed with secondary progressive MS (SPMS), and the remaining one with primary progressive MS. A female predominance was observed in the MS, NMOSD, and ODD groups (56.7%, 91.3%, and 78.3%, respectively). The mean onset age of the MS group was 30.9 ± 11.0 years, which was significantly younger than that of the other groups (*p* = 0.000). The mean disease duration and follow-up period were longer in the MS group than in the other groups (disease duration 76.5 ± 73.9 months, *p* = 0.004; follow-up period 101.8 ± 61.8 months, *p* = 0.006) (Table 1). The EDSS scores at the time of each MRI scan were higher in the NMOSD group, and these differences were statistically significant up to the second follow-up MRI scan (*p* = 0.000 and 0.001, respectively). Anti-AQP4 IgG antibody positivity was observed only in the NMOSD group, and anti-MOG antibody positivity was observed in one of eight (12.5%) patients with NMOSD and two of eight (25.0%) patients with ODD. Anti-Ro antibody positivity was significantly higher in the NMOSD group (28.6%) than in the other groups (*p* = 0.001). There was no statistically significant difference in the frequency of CSF OCB positivity and IgG index between the groups. There were no significant differences in initial presentation. The average number of MRI scans (2.9 ± 1.0) and the recurrence rate (56.7%) were significantly higher in the MS group than in the other groups (*p* = 0.000, 0.005).

### 3.2. Brain MRI Lesion Analysis

All patients underwent more than one brain MRI scan, ranging from one to four. Sixteen of 76 patients (21.1%) had IRLs. Of the 16 patients with IRLs, 14 were in the MS group (87.5%), and 2 were in the ODD group (12.5%) (*p* = 0.000). Of the 30 patients with MS, IRLs were observed in 14 (46.7%). Of the 14 patients, 13 were diagnosed with RRMS and the other was diagnosed with SPMS. The total number of IRLs was significantly higher in the MS group (0.8 ± 1.5, range 0–5) than in the NMOSD group (0.0 ± 0.0) and the ODD group (0.0 ± 0.2, range 0–1), with statistical significance *p* = 0.001. The location of IRLs in the MS group was 0.2 ± 0.4, 0.3 ± 0.7, 0.4 ± 0.8 in the cortical/juxtacortical, periventricular, and deep white matter areas, respectively (Table 2). The total number of WMLs was significantly higher in the MS group (17.2 ± 16.6) than in the other groups (NMOSD, 5.0 ± 11.6; ODD, 5.2 ± 9.5) (*p* = 0.000). The MS group showed a significantly higher ratio of IRLs/WMLs than did the other groups (*p* = 0.001). There was no significant difference between the groups in terms of confluent WMLs (*p* = 0.058). The enhanced lesions did not show a difference among groups; however, the number of T1-blackhole lesions was significantly higher in the MS group (2.2 ± 2.7, *p* = 0.000) compared with the other groups (Table 2).

### 3.3. Comparison of Clinical Characteristics According to the Presence of IRLs in MS

Patients with MS were divided into two groups based on IRL positivity: MS with IRLs (n = 14) and MS without IRLs (n = 16) (Table 3). There were no significant differences between the two groups in terms of sex, age at onset, disease duration, EDSS scores, CSF OCB, IgG index, or initial presentation. However, the EDSS scores of the MS with IRLs group tended to be higher than those of the MS without IRLs group for each MRI scan. There was no significant difference in the number of patients with recurrence: 10 of 14 (71.4%) patients in the MS with IRLs group and 7 of 16 (43.8%) patients in the MS without IRLs group. However, the average number of recurrences was 1.4 ± 1.2 in the MS with IRLs and 0.5±0.6 in the MS without IRLs groups, respectively (*p* = 0.031). There were no significant differences in acute rescue therapy; however, the frequency of second-line medication for preventive therapy was significantly higher in patients with MS with IRLs (50.0%) than in those without IRLs (13.3%) (*p* = 0.041). The treatment period did not differ between the groups (Table 3). Total WMLs were significantly higher in MS with IRLs (27.0 ± 16.8) than in MS without IRLs (8.6 ± 10.8) (*p* = 0.001). In contrast, the enhancing lesions and the T1-blackhole lesions did not show statistically significant differences between the groups (Table 3). There was no correlation between IRLs with T1-blackhole lesions in the MS group.

### 3.4. Dynamic Change in IRLs in MS

Among the 14 patients with IRLs in the MS group, 13, 9, and 6 had a second, third, and fourth MRI scan, respectively. During the follow-up period, the total dynamics of IRL counts increased in seven patients (50.0%), remained unchanged in four patients (28.6%), and decreased in two patients (14.3%). Over time, the proportion of patients with expanding IRLs decreased (Figure 1). The expansion of IRLs was mostly observed in the first to second (42.9%, 6 of 14 patients) and the second to third (28.6%, 4 of 14 patients) MRI scans. On the other hand, patients with decreased IRL dynamics were observed more at the third to fourth MRI scan than at the earlier scan.

### 3.5. Illustrative Case of Expanded IRLs

In June 2017, a 26-year-old male patient visited our hospital complaining of headaches and dizziness. MRI revealed several periventricular, brainstem, and spinal cord hyperintense T2-FLAIR lesions, some of which were active. The patient was diagnosed with MS and treated with intravenous corticosteroid 1 g pulse therapy for 5 days. After acute rescue therapy, his EDSS score was 0 (normal, no disability), and he received dimethyl fumarate as a maintenance treatment. However, in July 2019, he complained of headaches, dizziness, and general weakness, suggesting a clinical relapse (EDSS score 1, no disability, only minimal signs). The MRI showed interval evolution of T2 high-signal lesions in the medulla, brachium pontis, pons and periventricular white matter, and some enhancing lesions. Since then, he has received an intravenous natalizumab infusion every 4 weeks. However, a 1-year follow-up MRI scan showed newly noted T2 high-signal intensity lesions in the right temporal white matter, middle cerebral peduncle, and left parietal white matter, and some of them were enhanced after gadolinium injection. The patient was diagnosed with MS relapse, and we switched his maintenance treatment from natalizumab to cladribine. On the third MRI scan performed 2 years later, no clinical recurrence was observed, and new T2 lesions were observed (EDSS score, 1). Regarding the IRL dynamic change, four IRLs were observed in different locations on the first MRI scan in July 2019 (Figure 2). After the change to second-line medicine, a 1-year follow-up MRI scan showed that 2 IRLs remained in the original location, 2 IRLs had disappeared, and 3 new IRLs had appeared in another location. On the third MRI scan taken 2 years later, 4 old IRLs and new 3 IRLs were observed. Although there were no significant changes in clinical features and EDSS scores, an increase in T2 lesions and IRLs was observed on serial MRIs (Figure 2).

## 4. Discussion

In this study, 16 of the 76 patients (21.1%) had IRLs. Of the 16 patients with IRLs, 14 were in the MS group (87.5%), and 2 were in the ODD group (12.5%), indicating that IRLs were significantly specific for MS (*p* = 0.000). IRLs were observed in 14 (46.7%) of 30 patients in the MS group. Recently, a meta-analysis estimated that the pooled prevalence of IRLs was 40.6% in MS [6]. They analyzed 29 studies, of which 19 were conducted using 7T MRI. The mean disease duration ranged from 1.5 to 17 years in the meta-analysis. Notably, the prevalence of IRLs was significantly lower in studies involving patients with longer disease duration. In our study, despite using only 3T MRI, IRLs were observed more frequently than in the meta-analysis. This might be explained by the fact that the mean disease duration of MS in our study was 6.4 years, which was shorter than that reported in previous meta-analyses. In our study, the total number of IRLs and the ratio of total IRLs to total WMLs were significantly higher in the MS group than in the other groups. We analyzed the ratio of total IRLs to total WMLs to reduce the bias caused by more WMLs in MS. Because IRLs are lesions observed around pre-existing WMLs, as the number of WMLs increases, more IRLs may be observed. Under the concept of ratio, IRLs were also more frequently observed in the MS group than in the other groups in this study.

Many research studies have proposed that IRLs are specific imaging markers for MS. Clarke et al. reported that finding at least 1 lesion with IRLs achieved 100% specificity when used to differentiate clinically isolated syndrome (CIS) and non-MS groups. They also reported that IRLs appeared to be absent in other diseases such as NMOSD, Susac syndrome, and ischemic lesions of the CNS [5]. Another study reported by Jang et al. investigated the MRI findings of patients with MS and NMOSD, focusing on the presence of IRLs using 3T MRI, including the quantitative susceptibility mapping (QSM) technique. According to their research, the presence of at least one paramagnetic rim lesion showed good diagnostic performance in the differential diagnosis of MS [13]. Moreover, in the 2021 MAGNIMS-CMSC-NAIMS consensus recommendations, it was mentioned that IRLs have the potential to increase the MRI specificity in differentiating MS from other conditions [17]. Even though CVS has been suggested to be highly specific to MS and a useful imaging marker for diagnosing MS [2,18], previous studies have shown that the presence of CVS is not related to the clinical phenotype or disease progression in MS [2,3]. Therefore, IRLs have been highlighted because of their prognostic role in addition to their diagnostic role.

In the MS group, patients with IRLs had a significantly higher total number of WMLs, experienced more frequent recurrences, and were treated more frequently with second-line immunosuppressive agents than were patients without IRLs. The EDSS tended to be higher in the MS with IRLs group than in the MS without IRLs group, although this was without statistical significance. The presence of IRLs seems to reflect a more severe disease progression in MS. Although the results of recent studies on the relationship between IRLs and disease severity are controversial, some studies have proposed that IRLs are potential prognostic biomarkers. Absinta et al. reported that individuals with IRLs had a more aggressive disease (higher lesion load and reduced brain volume) and developed higher motor and cognitive disability or transitioned to disease progression at a younger age, despite treatment [12]. They classified patients into 3 groups: (1) 84 (44%) with no IRLs, (2) 66 (34%) with 1 to 3 IRLs, and (3) 42 (22%) with 4 or more IRLs. The prevalence of clinically progressive MS was 1.6-fold higher in individuals with 4 or more IRLs than in those without IRLs, and median EDSS scores were 1.5 (no IRLs), 1.5 (1 to 3 IRLs), and 2.5 (4 or more IRLs) in each group, respectively. In their neuropathological evaluation, chronic active lesions had a destructive core, and smoldering inflammation and demyelination were observed at the edge. However, as only a few independent studies have supported the association between IRLs and worse clinical outcomes in patients with MS, a large-scale meta-analysis of the prognostic role of IRLs is required.

In our study of the dynamic changes in IRLs, IRLs in the MS group were expanded in seven patients (50.0%), remained unchanged in four patients (28.6%), and decreased in two patients (14.3%) during the follow-up period. IRL expansion was mostly observed on the first to second and second to third MRI scans. These findings are consistent with the results of recent research showing that IRLs initially progressed in the disease course and diminished after approximately five years of its first appearance [19]. A long-term imaging study investigating the dynamic changes of IRLs using volumetric MRI sequences reported that WMLs with IRLs enlarged during the follow-up period, while WMLs without IRLs tended to shrink [11]. In another study, chronic lesions with and without rims had a mean CAGR (compound annual growth rate of adjusted log-lesion volumes) of 2.5% per year (expansion) and −4.7% per year (shrinkage), respectively [12]. In a pathological survey on 2476 WMLs in MS, expanding lesions were predominantly found in progressive MS and were suggested to indicate progressive disease activity [20]. On the other hand, lesion shrinkage might be related to tissue repair and reduced inflammatory activity with the resolution of extracellular water as part of inflammatory edema [21]. Having lesions with IRLs would be associated with the active disease process and severity. As volumetric MRI could not be performed in our study, we identified the expansion of IRLs by counting the number of IRLs instead of calculating the lesion volumes. In our illustrative case of MS, the total number of IRLs increased on three consecutive MRI scans. As the IRLs increased, the overall lesion load increased. During this period, the EDSS score of the patient increased from 0 to 1, and maintenance treatment was changed from primary to secondary immunosuppressive agents. Because of the relatively short disease duration (25 months), long-term dynamic changes in the IRLs, such as lesion shrinkage, were not observed in this patient. We analyzed T1-blackhole and T1-contrast-enhanced lesions in addition to IRLs and WMLs to identify the relationship between them and IRLs. Little research has been conducted on the relationship between IRLs, T1-blackhole, and T1-contrast-enhanced lesions. Absinta et al. reported that some rim lesions had initial contrast enhancement, which persisted after the enhancement resolved in 12 of 22 patients. Furthermore, compared with lesions with transient rims, those with persistent rims had more T1-hypointensity between 3 and 12 months, and patients with persistent rims had a poor prognosis [22]. However, since studies that show the relationship between T1-blackholes and IRLs are still lacking, further studies are required. Although our study showed no correlation with IRLs, T1-blackhole lesions were observed more frequently in the MS group than in other groups. T1-blackhole lesions are formed by tissue destruction of structural components and water influx, which expands the extracellular space [4]. Previous studies described the presence of T1-blackhole lesions as a characteristic of MS, and T1-blackholes are more common in patients with longer disease duration and more disease progression [4,23].

Our study had some limitations. First, this was a cross-sectional retrospective study conducted at a single center, which resulted in a small sample size and limited data collection. Second, we analyzed conventional 3T MRI scans. With the use of MRI scan sequences such as the 7T MRI scan or QSM protocol, the detection rate of IRLs has increased. Although imaging using 7T MRI is more sensitive than 3T MRI for detecting IRLs [5], a high IRL detection rate can be achieved even with conventional 3T MRI if optimized sequences are used. Because 7T MRI or QSM might not be widely available in clinical practice, we believe this point is our study’s strength. Third, the EDSS score did not significantly reflect disease severity in MS. Most patients were relatively well controlled, and their disease duration was shorter than that of patients in other studies, resulting in lower EDSS scores. There were few differences in the EDSS scores among the groups, which did not accurately reflect disease prognosis.

In conclusion, the assessment of iron accumulation has increased our understanding of the pathomechanism and disease progression of MS. IRLs are associated with chronic ongoing tissue destruction in MS and could represent a reliable specific imaging biomarker to improve the diagnosis of MS. In addition, IRLs appear to reflect more severe disease progression in MS. IRLs can be detected with conventional 3T MRI, which is widely available in clinical practice. Therefore, IRLs are expected to play a role as imaging biomarkers for MS in clinical practice. Further studies on the dynamic changes in IRLs on long-term follow-up images and the prognostic role of IRLs are required.

## Figures and Tables

**Figure 1 diagnostics-13-01866-f001:**
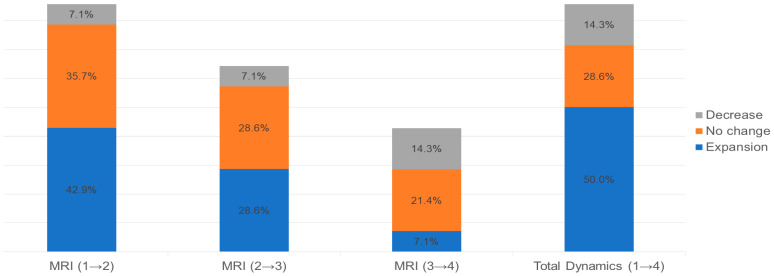
Dynamic changes in iron rim lesions (IRLs) in the multiple sclerosis (MS) group. In our study, of a total of 30 patients classified as the MS group, IRLs were observed in 14 patients. Overall, IRLs in the MS group were expanded in 7 patients (50.0%), unchanged in 4 patients (28.6%), and decreased in 2 patients (14.3%) during the follow-up period. The expansion of IRLs was mostly observed in the first to second (42.9%, 6 of 14 patients) and the second to third (28.6%, 4 of 14 patients) MRI scans. On the other hand, patients with decreased dynamics of IRLs were observed more at the third to fourth MRI scan than at the earlier scan. Thirteen patients could be analyzed in the MRI (1→2) column, 9 patients in the MRI (2→3) column, and 6 patients in the MRI (3→4) column.

**Figure 2 diagnostics-13-01866-f002:**
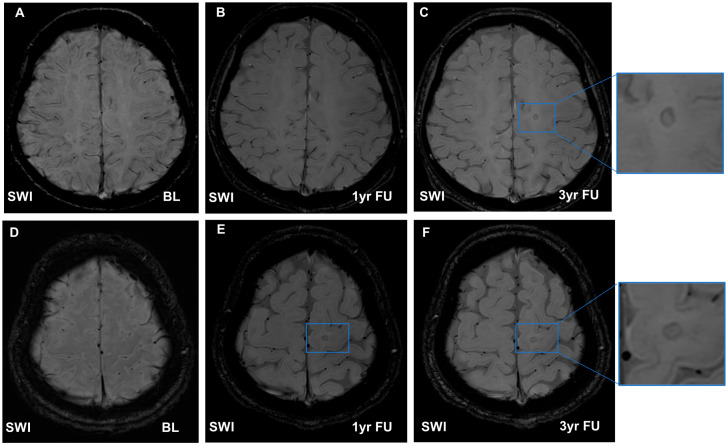
Serial brain magnetic resonance imaging of a 26-year-old male patient with relapsing-remitting multiple sclerosis. (**A**–**C**) are rims in susceptibility-weighted images (SWIs) of the same axial plane. There are no iron rim lesions (IRLs) in the baseline (**A**) and 1-year follow-up images (**B**). However, in the 3-year follow-up images, the IRLs are observed in the left frontal deep white matter area (**C**). (**D**–**F**) are SWIs of the same axial plane. There are no IRLs in the baseline (**D**) either, but (**E**,**F**) show the new IRLs in the left frontal subcortical white matter area.

**Table 1 diagnostics-13-01866-t001:** Clinical characteristics of the study population.

	MS (n = 30)	NMOSD (n = 23)	ODD (n = 23)	*p*-Value
Female (%)	17 (56.7%)	21 (91.3%)	18 (78.3%)	0.015 *
Onset age (years)	30.9 ± 11.0	45.8 ± 14.9	38.2 ± 12.9	0.000 *
Disease duration (months)	76.5 ± 73.9	52.5 ± 80.9	13.0 ± 29.1	0.004 *
Follow-up period (months)	101.8 ± 61.8	84.0 ± 85.8	42.7 ± 39.2	0.006 *
Number of MRIs	2.9 ± 1.0	1.4 ± 0.6	1.8 ± 1.0	0.000 *
EDSS on 1st MRI	1.0 ± 1.8 (n = 28)	4.1 ± 2.6 (n = 23)	1.4 ± 1.6 (n = 18)	0.000 *
EDSS on 2nd MRI	0.6 ± 1.3 (n = 27)	2.5 ± 1.0 (n = 7)	0.6 ± 0.7 (n = 9)	0.001 *
EDSS on 3rd MRI	0.6 ± 1.5 (n = 19)	1.0 (n = 1)	0.0 ± 0.0 (n = 5)	0.620
EDSS on 4th MRI	0.5 ± 0.9 (n = 11)	- (n = 0)	0.0 ± 0.0 (n = 2)	0.443
Aquaporin-4 antibodies (+) (%)	0/20 (0.0%)	20 (87.0%)	0/17 (0.0%)	0.000 *
MOG antibodies (+) (%)	0/2 (0.0%)	1/8 (12.5%)	2/8 (25.0%)	0.638
Anti-Ro antibodies (+) (%)	0/27 (0.0%)	6/21 (28.6%)	0/17 (0.0%)	0.001 *
Anti-La antibodies (+) (%)	0/27 (0.0%)	2/21 (9.5%)	0/17 (0.0%)	0.115
CSF OCB (+) (%)	10/23 (43.5%)	5/18 (27.8%)	2/18 (11.1%)	0.075
IgG index	1.0 ± 0.5 (0.49–2.47)	0.9 ± 1.3 (0.37–6.05)	0.5 ± 0.1 (0.28–0.84)	0.142
Initial presentation (%)				-
ON	8 (26.7%)	6 (26.1%)	5/22 (22.7%)	
TM	4 (13.3%)	12 (52.2%)	3/22 (13.6%)	
Cerebral Sx	10 (33.3%)	3 (13.0%)	6/22 (27.3%)	
Brainstem Sx	7 (23.3%)	2 (8.7%)	8/22 (36.4%)	
Myelitis + brain Sx	1 (3.3%)	0 (0.0%)	0/22 (0.0%)	
Recurrence (+) (%)	17 (56.7%)	10 (43.5%)	2 (13.0%)	0.005 *
Number of recurrences	1.0 ± 1.0	0.6 ± 0.7	0.1 ± 0.3	0.002 *

MS, multiple sclerosis; NMOSD, neuromyelitis optica spectrum disorders; ODD, other central nervous system demyelinating diseases, including optic neuritis; MRI, magnetic resonance imaging; EDSS, Expanded Disability Status Scale; MOG, myelin oligodendrocyte glycoprotein; CSF OCB, cerebrospinal fluid oligoclonal bands; ON, optic neuritis; TM, transverse myelitis; Sx, symptoms. - indicates that the result cannot be generated; * indicates that the results are statistically significant (*p* < 0.05).

**Table 2 diagnostics-13-01866-t002:** Comparison of first brain magnetic resonance imaging (MRI) scan findings in the study population.

Total (n = 76)	MS	NMOSD	ODD	*p*-Value
1st MRI scan	n = 30	n = 23	n = 23	
Total IRLs	0.8 ± 1.5 (0–5)	0.0 ± 0.0	0.0 ± 0.2 (0–1)	0.001 *
Cortical/Juxtacortical lesions	0.2 ± 0.4 (0–1)	0.0 ± 0.0	0.0 ± 0.0	0.007 *
Periventricular lesions	0.3 ± 0.7 (0–3)	0.0 ± 0.0	0.0 ± 0.2 (0–1)	0.062
Deep white matter lesions	0.4 ± 0.8 (0–3)	0.0 ± 0.0	0.0 ± 0.0	0.003 *
Total WMLs	17.2 ± 16.6	5.0 ± 11.6	4.1 ± 8.3	0.000 *
Periventricular lesions	4.0 ± 5.4	0.8 ± 1.6	0.5 ± 0.8	0.023 *
Ratio of IRLs/WMLs	0.06 ± 0.12 (0.00–0.50)	0.0 ± 0.0	0.01 ± 0.05 (0.00–0.25)	0.001 *
Confluent lesions (%)	11 (36.7%)	5 (21.7%)	2 (8.7%)	0.058
T1-contrast-enhanced lesions	0.2 ± 1.0	0.1 ± 0.3	0.1 ± 0.3	0.959
T1-blackhole lesions	2.2 ± 2.7	0.1 ± 0.5	0.5 ± 1.0	0.000 *

MRI, magnetic resonance imaging; MS, multiple sclerosis; NMOSD, neuromyelitis optica spectrum disorders; ODD, other central nervous system demyelinating diseases, including optic neuritis; IRLs, iron rim lesions; WMLs, white matter lesions. * indicates that the results are statistically significant (*p* < 0.05).

**Table 3 diagnostics-13-01866-t003:** Characteristics of multiple sclerosis (MS) subgroups according to the presence of iron rim lesions (IRLs).

Total (n = 30)	MS with IRLs (n = 14)	MS without IRLs (n = 16)	*p*-Value
Female (%)	6 (42.9%)	11 (68.8%)	0.269
Onset age (years)	27.3 ± 6.4	34.1 ± 13.2	0.080
Disease duration (months)	56.1 ± 46.5	94.3 ± 89.2	0.161
EDSS on 1st MRI	1.7 ± 2.6 (n = 12)	0.5 ± 0.8 (n =16)	0.223
EDSS on 2nd MRI	0.9 ± 1.8 (n = 13)	0.3 ± 0.5 (n = 14)	0.458
EDSS on 3rd MRI	1.2 ± 2.1 (n = 9)	0.1 ± 0.3 (n = 10)	0.079
EDSS on 4th MRI	0.8 ± 1.2 (n = 6)	0.2 ± 0.4 (n = 5)	0.429
CSF OCB (+) (%)	6/13 (46.2%)	4/10 (40.0%)	1.000
IgG index	1.1 ± 0.6 (0.51–2.47)	1.0 ± 0.5 (0.49–1.90)	0.804
Initial presentation (%)			0.187
ON	4 (28.6%)	4 (25.0%)	
TM	2 (14.3%)	2 (12.5%)	
Cerebral Sx	7 (50.0%)	3 (18.8%)	
Brainstem Sx	1 (7.1%)	6 (37.5%)	
Myelitis + brain Sx	0 (0.0%)	1 (6.3%)	
Number of MRIs	3.0 ± 1.0	2.8 ± 1.0	0.637
Recurrence (+) (%)	10 (71.4%)	7 (43.8%)	0.159
Number of recurrences	1.4 ± 1.2	0.5 ± 0.6	0.031 *
Acute rescue therapy (%)			0.483
None	1 (7.1%)	0/15 (0.0%)	
Intravenous corticosteroid	13 (92.9%)	15/15 (100.0%)	
Current preventive treatment (%)			0.041 *
1st line medication	7 (50.0%)	13/15 (86.7%)	
2nd line medication	7 (50.0%)	2/15 (13.3%)	
Treatment period (months)	82.6 ± 45.9	118.6 ± 70.1	0.113
1st MRI scan			
WMLs	27.0 ± 16.8 (2–50)	8.6 ± 10.8 (0–30)	0.001 *
T1-contrast-enhanced lesions	0.5 ± 1.4	0.1 ± 0.3	0.682
T1-blackhole lesions	2.8 ± 2.7	1.8 ± 2.6	0.319

MS, multiple sclerosis; IRLs, iron rim lesions; EDSS, Expanded Disability Status Scale; MRI, magnetic resonance imaging; CSF OCB, cerebrospinal fluid oligoclonal bands; ON, optic neuritis; TM, transverse myelitis; Sx, symptoms; WMLs, white matter lesions. * indicates that the results are statistically significant (*p* < 0.05).

## Data Availability

The data presented in this study are available on request from the corresponding author. The data are not publicly available due to privacy and ethical restrictions.

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
