# Peer review of "Iron Rim Lesions as a Specific and Prognostic Biomarker of Multiple Sclerosis: 3T-Based Susceptibility-Weighted Imaging"

_diagnostics, 2023, doi:10.3390/diagnostics13111866_

Round 1
Reviewer 1 Report
In this manuscript, the authors investigated the clinical significance of iron rim lesions (IRLs) detected with 3T MRI in distinguishing multiple sclerosis (MS) from other CNS demyelinating diseases. Patients with IRLs in the MS group had more white matter lesions, experienced frequent symptom recurrence, and received second-line immunosuppressive treatment more often. Although not novel, this study highlights the potential of IRLs as an imaging biomarker for improving MS diagnosis and partially reveals a potential prognostic value.
It is well-written, and well organized, but I have some comments to be addressed:
- Mean follow-up period ¿which is in each group and in MS?
- Frequency of MRI acquisition (3 months, 6 months, year...): is it comparable between groups? 1-2, 2-3,...
- EDSS change at the end of follow-up (or closure of the analysis) in both groups (w/ IRLs or not) corrected by follow-up time (Kaplan-Meier) although I understand that statistics with such low sample size should not be performed or might be misleading. I would like to know the data.
- You have not addressed the phenotype of MS patients. Were all RR, some SP or PP? was there a difference in IRLs?
Sincerely
Author Response
Manuscript number: diagnostics-2414330
Manuscript title: Irom rim lesions as a specific and prognostic biomarker of multiple sclerosis: 3T-based susceptibility-weighted imaging
Responses to the reviewer’s comments,
May 23, 2023
Dear reviewers and editorial staff from Diagnostics
We would like to thank the reviewers and the editor for their careful and thorough reading of this manuscript, and for their thoughtful comments and constructive suggestions, which helped to improve the manuscript. We have revised the manuscript to address all the issues raised by the reviewers. Our responses are as follows:
Reviewer 1
In this manuscript, the authors investigated the clinical significance of iron rim lesions (IRLs) detected with 3T MRI in distinguishing multiple sclerosis (MS) from other CNS demyelinating diseases. Patients with IRLs in the MS group had more white matter lesions, experienced frequent symptom recurrence, and received second-line immunosuppressive treatment more often. Although not novel, this study highlights the potential of IRLs as an imaging biomarker for improving MS diagnosis and partially reveals a potential prognostic value.
- Mean follow-up period which is in each group and in MS?
Answer) Mean follow-up period of each group is the following:
MS group, 101.8±61.8 (months); NMOSD group, 84.0±85.8; and ODD group, 42.7±39.2; respectively. Mean follow-up period was longer in the MS group than in other groups (P=0.006). We added that in Table 1 and described that at ‘Patient’s baseline characteristics’ of Results sections.
The mean disease duration was longer in the MS group than in the other groups (76.5±73.9 months, P=0.000) (Table 1).
-> The mean disease duration and follow-up period were longer in the MS group than in the other groups (disease duration 76.5±73.9 months, P=0.004; follow-up period 101.8±61.8 months, P=0.006) (Table 1).
Frequency of MRI acquisition (3 months, 6 months, year...): is it comparable between groups? 1-2, 2-3,...
Answer) We entirely agree with the reviewer’s comment. A comparison of serial MRI in patients with MS was shown in Figure 1, and that was in-group analysis. MS patients were mostly followed up with MRI annually. In addition to MS group, the frequency of MRI acquisition is comparable between each group in this study. Mean duration of serial MRI acquisition in MS group was about 12 months. In NMOSD group, mean duration for first to second MRI scan was 10.3±6.0 months, and that for second to third MRI scan was 10.0 months. There were no patients who performed fourth scan of MRI in NMOSD group. In ODD group, mean duration for first to second MRI scan was 10.6±4.5 months, that for second to third MRI scan was 12.8±1.5 months, and that for third to fourth MRI scan was 12.5±0.7 months. And there were no significant differences of mean duration for MRI scan among the groups (mean duration for first to second MR scan, P=0.317; mean duration for second to third MR scan P=0.605; and mean duration for third to fourth MR scan P=0.463). If the reviewer mentions that it is necessary to add the comparison of mean duration for each MR scan among the groups to the table, I will.
- EDSS change at the end of follow-up (or closure of the analysis) in both groups (w/ IRLs or not) corrected by follow-up time (Kaplan-Meier) although I understand that statistics with such low sample size should not be performed or might be misleading. I would like to know the data.
Answer) We entirely agree with the reviewer’s comment. We checked EDSS score whenever MRI was performed in both groups (MS with IRL, and MS without IRL), and compared EDSS score between the two groups each time. As the reviewer mentioned, because the sample size was small and it did not follow a normal distribution, the analysis was performed by Mann-Whitney U test as a distribution-free method (nonparametric test).
You have not addressed the phenotype of MS patients. Were all RR, some SP or PP? was there a difference in IRLs?
Answer) Thank you for your comments. Of the 30 MS patients, 28 were RRMS, one was diagnosed with SPMS and the remaining one with PPMS. Although the in-group comparison was not possible due to the small number of SPMS and PPMS patients, IRL was not observed in patients diagnosed with PPMS, and IRL was observed in patients diagnosed with SPMS. In patient with SPMS, the IRL was observed as 5 on the first MRI and 9 on the second and third MRI, showing the pattern of expansion of the IRL. This was additionally described in revised manuscript.
In ‘Patient’s baseline characteristics’ of Result section
A total of 76 patients were included in the study (MS, 30; NMOSD, 23; and ODD, 23). Of the 30 MS patients, 28 were relapse-remitting MS (RRMS), one was diagnosed with secondary progressive MS (SPMS), and the remaining one with primary progressive MS.
In ‘Brain MRI lesion analysis’ of Result section
Of the 16 patients with IRLs, 14 were in the MS group (87.5%), and two were in the ODD group (12.5%) (P=0.000). Of the 30 patients with MS, IRLs were observed in 14 (46.7%). Of the 14 patients, 13 were diagnosed with RRMS and the other was diagnosed with SPMS.
We hope the revised manuscript will better meet the requirements of Diagnostics for publication. We thank you once again for the delicate and constructive review provided by editors and referees.

Reviewer 2 Report
The paper is well written and covers a very interesting topic of current clinical interest. But here are some questions to clear:
1. Figure 1 - I cannot understand the columns %, why summary % do not match in all the columns? And the summary of % in each column in not 100% = 14 patients. What the rest of % do not mentioned in the columns mean?
2. Figure 2 - the SWI protocol for picture A and D seems to differ from the rest of MRI scans. So here comes the question about SWI protocols comparability. Is there possibility that iron could be better seen in one SWI protocol better than in other? So some IRLs could be just elusive? If not, probably it s worth mentioning some words about protocols comparability.
3. It s not clear why authors decided to include transverse myelitis group if this group data wasn t described in results section. Probably it s better to cut this group from the article at all? The diagnosis "transverse myelitis" implies that there s no brain lesions.
Author Response
Manuscript number: diagnostics-2414330
Manuscript title: Irom rim lesions as a specific and prognostic biomarker of multiple sclerosis: 3T-based susceptibility-weighted imaging
Responses to the reviewer’s comments,
May 23, 2023
Dear reviewers and editorial staff from Diagnostics
We would like to thank the reviewers and the editor for their careful and thorough reading of this manuscript, and for their thoughtful comments and constructive suggestions, which helped to improve the manuscript. We have revised the manuscript to address all the issues raised by the reviewers. Our responses are as follows:
Reviewer 2
The paper is well written and covers a very interesting topic of current clinical interest. But here are some questions to clear:
- Figure 1 - I cannot understand the columns %, why summary % do not match in all the columns? And the summary of % in each column in not 100% = 14 patients. What the rest of % do not mentioned in the columns mean?
Answer) Thank you for your delicate and thoughtful comment. There were 14 MS patients with IRLs, and 1 of them did not perform a second MRI, so the total number of patients available for analysis in the 1à2 MR scan column was 13. In the 1->2 MR scan column, expansion of IRLs was 6 out of 13, no change was 5 out of 13, and decrease was 1 out of 13, and the column showed a proportion of total 14 patients (42.9% [6/14], 35.7% [5/14], and 7.1% [1/14]). And 9 patients could be analyzed in 2-> 3 MR scan, of which 4 were expansion, 4 were no change, and 1 were decrease of IRLs (28.6% [4/14], 28.6% [4/14], and 7.1% [1/14]). Six patients could be analyzed in 3->4 MR scan, of which 1 was expansion, 2 was no change, and 3 were decrease of IRLs. To understand this, the legend of Figure 1 and manuscript were revised.
In Figure 1
The expansion of IRLs was mostly observed in first to second (42.9%, 6 of 14 patients) and second to third (28.6%, 4 of 14 patients) scans of MRI. On the other hand, patients with decreased dynamics of IRLs were observed more at the third to fourth scan of MRI than at the earlier scan. *Thirteen patients could be analyzed in MRI (1->2) column, 9 patients in MRI (2->3) column, and 6 patients in MRI (3->4) column.
‘Dynamic change of IRLs in MS’ of Result section
Among the 14 patients with IRL in MS group, 13, 9, and 6 had a second, third, and fourth MRI scan, respectively. During the follow-up period, the total dynamics of IRLs counts increased in seven patients (50.0%), remained unchanged in four patients (28.6%), and decreased in two patients (14.3%). Over time, the proportion of patients with expanding IRLs decreased (Figure 1). The expansion of IRLs was mostly observed in first to second (42.9%, 6 of 14 patients) and second to third (28.6%, 4 of 14 patients) scans of MRI. On the other hand, patients with decreased dynamics of IRLs were observed more at the third to fourth scan of MRI than at the earlier scan.
- Figure 2 - the SWI protocol for picture A and D seems to differ from the rest of MRI scans. So here comes the question about SWI protocols comparability. Is there possibility that iron could be better seen in one SWI protocol better than in other? So some IRLs could be just elusive? If not, probably it s worth mentioning some words about protocols comparability.
Answer) Typical three SWI image protocols are followings:
Magnitude image, high-filtered phase image, and SWI (combined post-processed magnitude and phase images). Often a fourth set of images is provided, minimum intensity projection (mIP) image which is a thick slab of the conventional SWI images and is better able to demonstrate venous anatomy. Our center began to establish the SWI protocol for diagnosing MS from 2019. The first MRI of the case patient was performed before new protocol establishment, and there are only SWI and mIP images. The second and third MRI of the patient were performed after new protocol establishment, and it include magnitude image, high-filtered phase image, and mIP image. We compared the SWI image of 2019 with the magnitude image of 2020, and 2022. Since SWI is an image generated by combining magnitude and phase image, they can be compared with each other. Also, the parameters of all serial MRI such as long echo times (TE), flip angles (FA), and TR were the same in case patient. We reviewed the magnitude, phase image, and mIP images of all enrolled patients to confirm the presence of IRLs. We described it in our manuscript in ‘MRI evaluation’ of Materials & method section.
- It s not clear why authors decided to include transverse myelitis group if this group data wasn t described in results section. Probably it s better to cut this group from the article at all? The diagnosis "transverse myelitis" implies that there s no brain lesions.
Answer) We entirely agree with the reviewer’s comment. As the reviewer mentioned, the data of TM need to be deleted in this study, so the manuscript was overall revised.
Details are shown in track version of revised manuscript.
We hope the revised manuscript will better meet the requirements of Diagnostics for publication. We thank you once again for the delicate and constructive review provided by editors and referees.
